# TACO: Vision Models Can Be Efficiently Specialized via Few-Shot Task-Aware Compression

## Abstract

Recent vision architectures and self-supervised training methods have enabled training computer vision models that are extremely accurate, but come with massive computational costs. In settings such as identifying species in camera traps in the field, users have limited resources, and may fine-tune a pretrained model on (often limited) data from a small set of specific categories of interest. Such users may still wish to make use of highly-accurate large models, but are often constrained by the computational cost. To address this, we ask: **can we quickly compress generalist models into accurate and efficient specialists given a small amount of data?**

Towards this goal, we propose a simple and versatile technique, which we call *Few-Shot **T**ask-**A**ware **CO**mpression (TACO)*. Given a general-purpose model pretrained on a broad task, such as classification on ImageNet or iNaturalist datasets with thousands of categories, TACO produces a much smaller model that is accurate on specialized tasks, such as classifying across vehicle types or animal species, based only on a few examples from each target class. The method is based on two key insights - 1) a powerful specialization effect for data-aware compression, which we showcase for the first time; 2) a dedicated fine-tuning procedure with knowledge distillation, which prevents overfitting even in scenarios where data is very scarce. Specifically, TACO is applied in *few-shot fashion*, i.e. only a few task-specific samples are used for compression, and the procedure has low computational overhead. We validate this approach experimentally using highly-accurate ResNet, ViT/DeiT, and ConvNeXt models, originally trained on ImageNet and iNaturalist datasets, which we specialize and compress to a diverse set of "downstream" subtasks, with notable computational speedups on both CPU and GPU.

## 1 Introduction

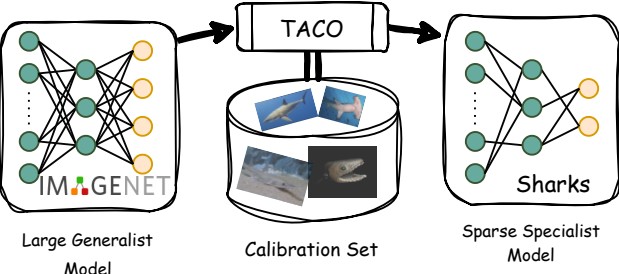

Figure 1: Few-Shot **T**ask-**A**ware **CO**mpression (TACO) is done using a calibration set, which is simply a few-shot subset of the desired task. It consists of a single-step compression procedure, followed by a few-shot finetuning step.

The recent introduction of new training techniques that can leverage massive amounts of unlabelled data, such as self-supervised (contrastive) learning (Chen et al., 2020; Radford et al., 2021), complemented by new

architectures, such as Vision Transformers (ViTs) (Dosovitskiy et al., 2020; Touvron et al., 2021) and next-generation convolutional networks (e.g., ConvNeXt) (Liu et al., 2022), has led to unprecedented accuracy on standard benchmarks, such as the ImageNet image classification task, even in zero- or few-shot settings.

Yet, this impressive increase in accuracy comes at the price of large parameter counts and computational costs. For example, the highly accurate "large" variants of the Vision Transformer (ViT/DeiT) model family, commonly used for pretraining, can easily count more than 200 Million parameters, requiring more than 50 GFLOPs for inference at standard input sizes. Thus, although these models reach extremely high accuracies for image classification—e.g., close to 90% Top-1 accuracy on ImageNet-1K—they are impractical to deploy, especially in resource-constrained settings, such as edge inference.

One popular way of bridging this computational divide is *post-training compression*, e.g. Nagel et al. (2020); Hubara et al. (2021); Frantar et al. (2022): given a small but representative calibration dataset, the model is compressed to a given sparsity or low-precision requirement *in one compression step*. However, highly-accurate models, such as the ones from the Vision Transformer family, are notoriously hard to compress Chen et al. (2021a): intuitively, these models may be inherently parameter-heavy, as they are *generalists*, trained across massive and often noisy datasets, and classifying images across thousands of output classes.

In this paper, we seek to extend the reach of post-training compression to consider the requirement of *model specialization*. We start from the idea that, in practical deployment scenarios, such as an ecologist seeking to identify animal species in camera trap images (Beery et al., 2018) or a city planner seeking to identify vehicles using a traffic camera (Ozkurt & Camci, 2009), users only need their models to categorize a *smaller set of classes of interest*. Thus, we shift the objective to producing *efficient specialist models*, which maintain high accuracy on a narrower subset of the data distribution. Specialization should allow for higher compression, as only a fraction of the features should be required over narrower tasks.

We propose a simple and versatile approach which we call **T**ask-**A**ware **CO**mpression (TACO), illustrated in Figure 1. Given a *pre-trained generalist model* and a "calibration set" containing a few samples for categories of interest, TACO can produce an efficient specialist model in two steps. The first is a *task-aware compression step*, which reduces the model size layer-wise, preserving the model's features on the task-specific calibration data, but removing unnecessary parameters. Here, our key insight is that existing highly-accurate compression solvers can also be leveraged for *model specialization* over a task-specific calibration set. The second step is a *few-shot tuning* procedure, where we finetune the compressed model on the same small calibration set, leveraging information from the uncompressed model to prevent overfitting.

We validate TACO experimentally on a wide range of pretraining settings, model architectures, and target tasks. Specifically, we start from models trained on diverse and general tasks such as ImageNet ILSVRC12 (Russakovsky et al., 2015) and iNaturalist 2021 Van Horn et al. (2018). We consider highly-accurate Base-size models on these datasets from the Vision Transformer (ViT), Swin (Liu et al., 2021) and ConvNeXt families, which we specialize across tasks obtained by restricting the taxonomies to natural subsets such as animal species or vehicle classes. We also extend TACO to a standard transfer learning setting, showing that it can produce accurate data-aware "lottery tickets" (Chen et al., 2021b). We compress models with both unstructured and structured pruning, and examine model specialization, as well as practical speedups due to compression. In addition, we also showcase TACO on the popular `diffusers` text-to-image generative task (Hugging Face, Inc., 2023).

Our results show that TACO can reduce the number of model parameters by 10-20x, with low to moderate accuracy loss on the target task. We show that this compression is achieved via a *task specialization effect*: TACO models have significantly better accuracy at the same compression level than models compressed via standard post-training compression. As a rule of thumb, TACO can double the degree of sparsity, at the same accuracy level on the target task, relative to generic post-training compression.

The TACO procedure is computationally efficient: for instance, we can specialize an 86M parameter ViT-Base model in ∼10 minutes on a single GPU. For instance, this allows us to specialize over a large fraction of the WordNet taxonomy (166 tasks) in reasonable time. Moreover, TACO supports different solvers for imposing sparsity: we instantiate it to structured and unstructured pruning, and propose a "hybrid" solver, which is both accurate and highly-efficient. For practical gains, TACO models provide end-to-end

computational speedups 1.5-3x relative to the uncompressed models, at similar accuracy levels to dense, and 3-6x at moderate accuracy loss, on both CPUs and GPUs.

**Related work.** *Few-shot learning* (Song et al., 2022; Parnami & Lee, 2022) is a task of adapting a pre-trained generalist model to a limited number of labeled examples. In many cases, a large amount of data is not available due to the cost of data collection or privacy concerns. Therefore, the ability of the model to adapt to a task of interest given a small dataset is of significant importance. Similarly to work on few-shot learning, we adopt a few samples for each class. However, the problem addressed in this work is different. Few-shot learning tackles the problem of data-efficient model adaptation, i.e., learning a new ability. On the contrary, we address the problem of preserving specific knowledge *already present in the model.*

## 2 Method

TACO involves two stages: a task-aware compression step, and a few-shot tuning step. Both steps utilize *the same* small representative set of $k$ samples per class corresponding to the task of interest (say $k = 10$).

### 2.1 Few-Shot Task-Aware COmpression

The first step of TACO performs one-shot compression on the given model and task, reducing the model size in a layer-wise fashion by leveraging a set of task-specific calibration data $\mathcal{D}$. At a high level, given an uncompressed model $f$, a sparsity solver $S$, task-specific calibration data $\mathcal{D}$, and compression constraint $\mathcal{C}$, the TACO procedure produces a compressed version

$$f_{\mathcal{C}}^{\mathcal{D}} = \text{TACO}(f, S, \mathcal{C}, \mathcal{D}).$$

To implement this, we follow the post-training compression approach: we split the network layer-wise, and run task-specific calibration data $\mathcal{D}$ through the uncompressed model. For each network layer $\ell \in \{1, 2, \ldots, L\}$ with weights $\mathbf{W}_\ell$, we extract the input $\mathbf{X}_\ell^{\mathcal{D}}$ on the calibration data $\mathcal{D}$. Compression is then applied to each layer by solving a layer-wise optimization problem, i.e. identifying the layer weights $\widehat{\mathbf{W}}$ which minimize the difference between the dense and compressed layer output on the provided calibration data, measured in L2-norm. For example, for a linear layer, our goal is to determine a set of weights $\widehat{\mathbf{W}}_\ell^\star$ satisfying a compression predicate $\mathcal{C}$ such that:

$$\widehat{\mathbf{W}}_\ell^\star = \underset{\widehat{\mathbf{w}}_\ell \in \mathcal{C}}{\arg\min} \|\widehat{\mathbf{W}}_\ell \mathbf{X}_\ell^{\mathcal{D}} - \mathbf{W}_\ell \mathbf{X}_\ell^{\mathcal{D}}\|_2^2. \tag{1}$$

The key difference relative to prior work on post-training compression is *specialization over task-aware calibration data*: While prior work uses *generic* calibration data, our hypothesis is that, when applied over task-specific calibration data, a good solver for this constrained optimization problem can also *isolate the features or parameters* that are specific to the specialized task. This should be done by considering the task-specific calibration data during the solving, as the weights alone are initially generalists.

#### 2.1.1 Solving the Task-Aware Compression Problem

**Sparsity Solvers.** To test the specialization hypothesis, we investigated the three existing solvers for unstructured sparsity in the post-training setting:

1. *Magnitude Pruning* (Han et al., 2015), which sorts the weights by their absolute value, is *data-agnostic*, so we use it as a reasonable baseline.

2. The *Optimal Brain Compression (OBC)* solver (Frantar et al., 2022) approximates an optimal solution for the problem in Equation 1 by greedily pruning the weight with the lowest impact on the layer's L2 loss, and updating all remaining weights to best compensate for the incurred compression error. The solver's asymptotic complexity is $\mathcal{O}(d_{row} \cdot d_{col}^3)$, where $d_{row}$ is the output dimension of the layer and $d_{col}$ is either the number of input features (for a linear layer) or the number of channels multiplied by kernel size for a convolutional layer. This is the most accurate known solver for layer-wise sparsity but can take several hours to apply to a model with more than 100M parameters.

3. The *FastOBC (SparseGPT)* (Frantar & Alistarh, 2023) solver approximates the OBC approach by pruning in a computationally-efficient block pattern. The solver is $> 100$ x faster than OBC for the same layer size, at the cost of lower accuracy in the provided solution at a given sparsity.

**A Hybrid Solver.** We also propose *HybridOBC*, a solver which can be seen as a best-of-both-worlds composition of OBC and FastOBC. Most of the runtime cost of OBC comes from the layers with the largest input dimensions, as its complexity is $\mathcal{O}(d_{row} \cdot d_{col}^3)$. Therefore, we prune sensitive layers, with smaller input dimensions, via the accurate OBC pruner, and the ones with large values of $d_{col}$ (for example, `fc2` layer in ViTs) with FastOBC. We will show that HybridOBC significantly reduces the runtime cost relative to OBC, with negligible loss of accuracy.

**Alternative Solvers.** We have also considered sparsity solvers that leverage the Fisher approximation, such as WoodFisher (Singh & Alistarh, 2020; Kurtic et al., 2022) or CAP (Kuznedelev et al., 2024). In our experiments, these performed worse in terms of accuracy, relative to the $\ell_2$ layer-wise pruners above in the low-data regime we consider in the paper, i.e. 10 samples per class. This is natural since the Fisher approximation requires a large number of gradient samples to be accurate. However, as we show in Section 3.5, if the data is more abundant (i.e., 100 samples/class), one can obtain even higher accuracy, especially at high sparsities. In addition, we also consider the *ZipLM* (Kurtic et al., 2023) solver for *structured* (row-wise) sparsity. We observe a task specialization effect in this case as well.

## 2.2 Few-Shot Task-Aware Tuning

At high compression rates, e.g. $>90\%$ sparsity, the above one-shot layer-wise approach can lead to significant accuracy degradation. Therefore, we employ a limited few-shot form of finetuning to re-calibrate the model and recover additional accuracy. Yet, naive finetuning on a small amount of data results in "overfitting:" the calibration samples are memorized, but validation performance does not improve. In addition, *sparse* models tend to be more difficult to optimize (Evci et al., 2019). We address both challenges by introducing a self-distillation-based approach (Zhang et al., 2019; 2021) to recover the accuracy of the sparsified model relative to the dense one, which does not overfit when applied over a small calibration set.

Recall that, in standard distillation, the student network is trained to reproduce the distribution of the teacher model via minimization of the KL-divergence:

$$\mathcal{L}_{\text{logit}} = D_{\text{KL}}(\theta_t || \theta_s) = \frac{1}{B} \sum_{i=1}^{B} p_{\theta_t}(\mathbf{x}_i) \log \frac{p_{\theta_t}(\mathbf{x}_i)}{p_{\theta_s}(\mathbf{x}_i)} \tag{2}$$

However, standard distillation may not provide sufficient information in the case of extremely little data. Instead, we found it beneficial to transfer intermediate activations between the dense model and the sparse one. Specifically, we apply normalized mean squared error loss (NMSE) between teacher and student features at the output of each residual block:

$$\mathcal{L}_{\text{feat}}^l = \frac{\text{MSE}(f_t^l, f_s^l)}{\text{MSE}(f_t^l, 0)} \tag{3}$$

Above $f_t^l$ and $f_s^l$ denote teacher and student features, (that is, 3-dimensional feature maps for ViT models and 4-dimensional for ConvNets) at the output of layer $l$ and MSE represents the mean squared error calculated as $\text{MSE}(X, Y) = \frac{1}{N} \sum_{i=0}^{N} (x_i - y_i)^2$, for $N$-dimensional vectors $X$ and $Y$. Normalization factor accounts for different magnitude of activations across layers of the model. This form of feature-level distillation significantly improves performance, relative to both standard retraining and regular distillation (Figure 8 (**left**)). We note, that variants of this loss were used in prior work (Sun et al., 2019; Kurtic et al., 2023; Frantar & Alistarh, 2022). The resulting total loss function has the following form:

$$\mathcal{L}_{total} = \mathcal{L}_{task} + \mathcal{L}_{logit} + \mathcal{L}_{feat} \tag{4}$$

The relative weighting of losses can be tuned, but we found uniform weighting to work well (see Figure 14).

# 3    Experiments

## 3.1    Experimental setup

We compare Task-Aware COmpression, where the objective in Equation equation 1 is optimized on task-specific data, with task-agnostic *Post-Training Compression* (PTC), where the data is chosen uniformly at random over a general dataset such as ImageNet-1k and iNaturalist21, using the same number of samples for both methods.

The impact of task specialization is assessed for *one-shot* compression, where accuracy without further tuning is reported, and for *fine-tuned accuracy*, where we assess the model performance after fine-tuning in few-shot.

In the one-shot setup, one can directly compare the impact of the choice of calibration data on the solution found by the sparse solver. In the fine-tuning setting, we examine whether the accuracy difference between PTC and TACO vanishes after tuning or whether fine-tuning serves to bridge the performance gap. In all experiments below we compare PTC and TACO after fine-tuning, and additionally consider results after only single-step compression for ImageNet subtasks.

**Generalist Datasets.**    We consider two large image classification benchmarks: ImageNet ILSVRC12 Russakovsky et al. (2015), and iNaturalist 2021 Van Horn et al. (2018). Models trained on ImageNet-1k are used in transfer learning experiments.

**Generalist models.**    In the experiments below, we considered three models pretrained on ImageNet-1k, specifically, DeiT-III-Base following Touvron et al. (2022), the Transformer-CNN hybrid Swin model (Liu et al., 2021) and the modern Transformer-inspired CNNs (ConvNeXt) (Liu et al., 2022). For iNaturalist experiments, we adopted ViT-Base from  He et al. (2022) and finetuned it on iNaturalist using the same hyperparameters as He et al. (2022). All models are roughly of the same size $\sim 90M$ parameters.

**Task Specialization.**    For ImageNet, we consider specialization across subtasks of the WordNet hierarchy corresponding to ImageNet-1K, generated via the `robustness` library (Engstrom et al., 2019). We selected subtasks that contain at least 5 classes: this totals 166 possible subtasks on ImageNet. For iNaturalist, we specialize on various branches of the taxonomic tree corresponding to the phylum, family, order, and genus levels, in increasing order of specificity.

**Training parameters and the effect of data scaling.**    In the experiments below, **we use 10 samples per class** for all tasks considered. Models are fine-tuned over 100 passes through the calibration data. Although this may seem a high number, we emphasize that the calibration set is tiny, and typically a whole pass involves a few dozen batches. The finetuning procedure takes $\sim$10 minutes on a single NVIDIA RTX 3090 GPU. In our experiments, we use a batch size of 16 for ImageNet and transfer learning subtasks (see Section 3.7) and 32 for iNaturalist subtasks. As expected, more data and longer training improve performance of the model. Ablation w.r.t. the samples per class and calibration passes are given in Figure 9.

**Details of the pruning setup.**    As standard (Gale et al., 2019), we do not prune input embeddings and the classification head, since their latency cost is small. All other linear layers and convolutions are pruned to same *uniform sparsity* level.

## 3.2    The TACO Effect

To motivate the need for task-specific compression we first apply the strategy introduced above to the compression on whole ImageNet-1k dataset, shown in Figure 2. The performance drop is quite significant even at moderate sparsity (e.g., 5% relative at 70% sparsity). For high sparsity (90%), the accuracy drops are extremely significant ($\geq 20\%$).

It is intuitive that one could achieve better performance for the same compression rate on a more specialized task. To test this hypothesis, we considered a few subtasks from ImageNet taxonomy: `vehicle`, `dog` and

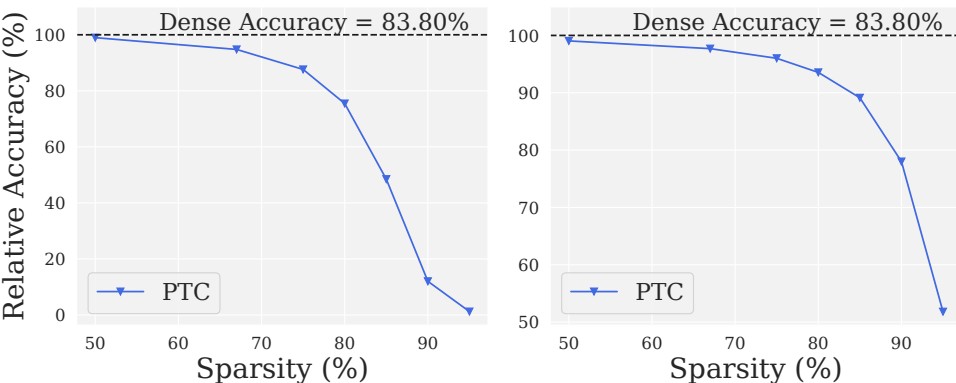

Figure 2: (**Left**) Single-step compression on ImageNet-1k. (**Right**) Finetuning on ImageNet-1k.

`spider` subtasks with 21, 116, and 6 sub-classes, respectively. The experiments below are performed with the DeiT-III-Base model, unless otherwise specified.

Specifically, the generalist pruning algorithm seeks to optimize performance over all 1K ImageNet classes, whereas the task-aware TACO sparsity solver optimizes the performance *only* on specific categories of interest.

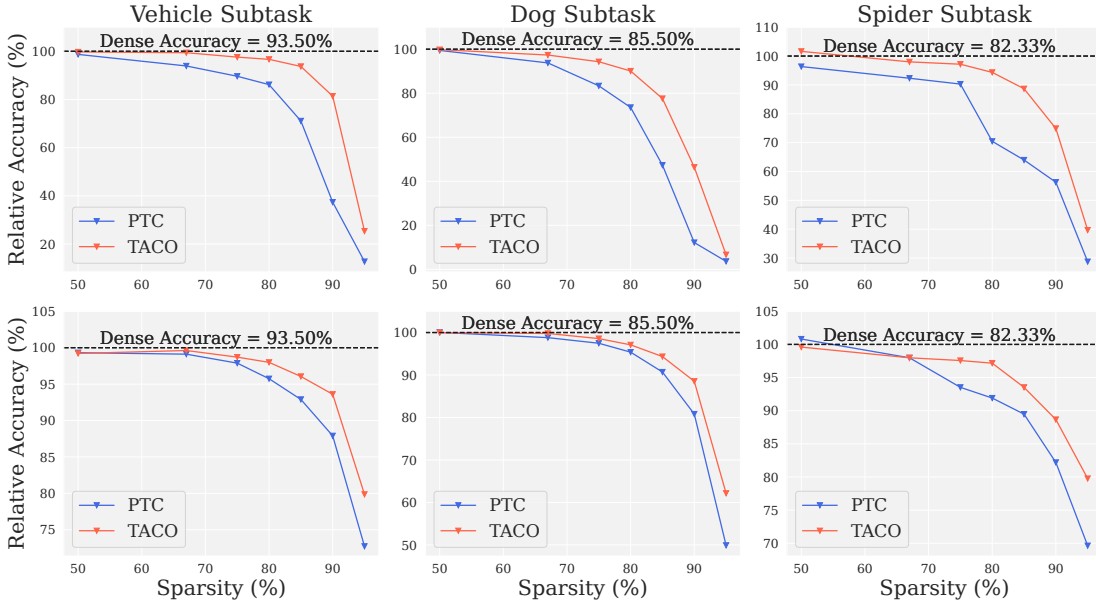

Figure 3: (**Top**) Single-step compression on ImageNet subtasks. (**Bottom**) Finetuning on ImageNet substasks.

We can observe in Figure 3 that the sparse solution found by TACO is noticeably better than the one produced by task-agnostic PTC, showcasing the "TACO effect" by which data-aware compression imposes task specialization in the resulting model. Importantly, the performance gap between TACO and PTC remains significant even after fine-tuning with distillation for both methods. This shows that the model specialization is preserved after fine-tuning. Specifically, one can compress the model to 75% sparsity with a minor drop in accuracy, and to 90% sparsity while preserving 90% of the dense model accuracy.

### 3.2.1 Task Complexity and Model Compressibility

**Task Complexity.** In the next experiment, we wish to examine compressibility with respect to task complexity, measured as the number of task sub-classes. For this, we run one-shot pruning over ImageNet

with respect to the WordNet hierarchy (Fellbaum, 2010). We keep only the tasks containing at least 5 output classes. We select three sparsity levels (0.6, 0.7, and 0.8) and obtain single-step pruned models using TACO. (We use the FastOBC solver in this experiment, given the large number–166–of subsets.) The results of this large-scale experiment are provided in Figure 4, where we examine the correlation between the accuracy of the sparse models relative to the corresponding dense one, as well as the relative accuracy drop post-compression, relative to the number of classes in the target specialization dataset.

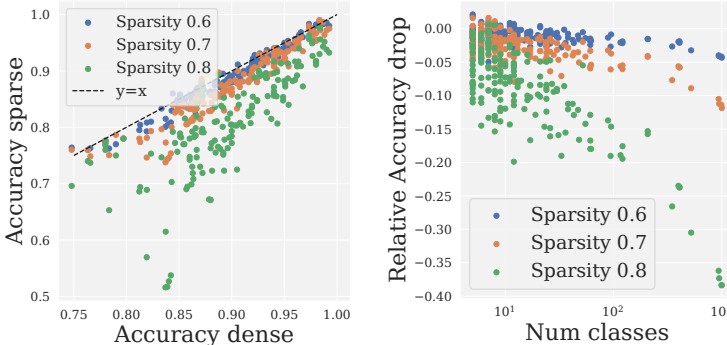

Figure 4: (**Left**) Accuracy on the ImageNet subtask for the dense model vs. accuracy of the sparse model. (**Right**) Relative accuracy drop vs. the number of classes.

In the left-hand plot, we observe a correlation between the accuracy of the uncompressed model and that of the sparse one on the given task. For most tasks, the performance drop is negligible for 60-70% sparsity and becomes pronounced at only 80% sparsity. Second, certain subtasks allow for higher compression relative to others. This correlates with the number of target classes, as a measure of "complexity." Specifically, the accuracy drop at a given sparsity appears to be *logarithmic* in the number of classes. Finally, for some subsets, the accuracy of the compressed model can be superior to the original model.

In addition, for each subtask selected, we evaluate the PTC model and compare the performance of PTC and TACO. Due to space constraints, results are presented in the Appendix F; overall, they show that TACO outperforms PTC on the majority of the tasks, and that the accuracy difference becomes more pronounced with an increase in sparsity. The difference is also more significant for smaller subtasks, showing the advantage of task specialization.

### 3.2.2   Additional Models and Structured Sparsity

Our conclusions generalize to other vision models and to structured sparsity. Specifically, we apply the ZipLM (Kurtic et al., 2023) structured solver, which optimizes a layer-wise objective and identifies the least important channels. Following Kurtic et al. (2023), we prune both attention heads and hidden dimensions in the feedforward Transformer layers, to the same sparsity level. The advantage of this approach is that structured sparsity can be accelerated both on CPU and GPU hardware, although we would expect higher accuracy drops.

Figure 5 (left) shows that we can prune the model to 30-50% sparsity with minor performance drops, and to 70% sparsity while preserving 90% of the original accuracy. Throughout, TACO outperforms PTC across sparsities, similarly to the unstructured case.

In Figure 5 (right), we evaluate TACO on two other recent vision models, ConvNext-Base and Swin-V2-Base, roughly of the same size and performance as DeiT-III-Base. The trend is similar across all models considered - up to 80-85% sparsity, performance drops are small. DeiT-III-Base appears to be the easiest to compress.

### 3.3   TACO on iNaturalist

We further investigate task-aware compression on the challenging iNaturalist21 dataset Van Horn et al. (2018), comprising 10,000 classes corresponding to diverse terrestrial and marine animal and plant species.

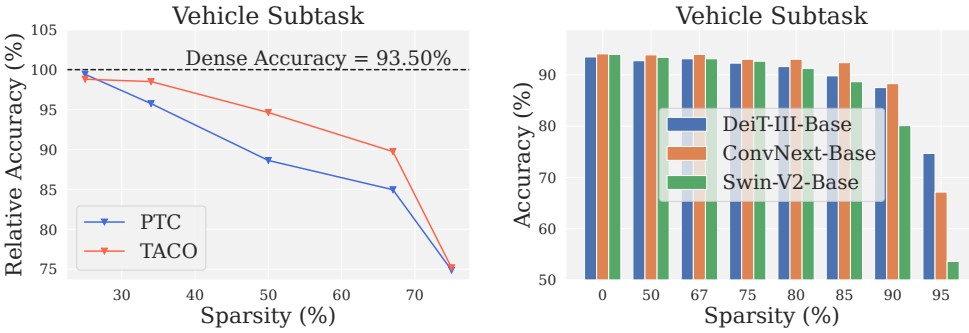

Figure 5: (**Left**) Structured pruning on "Vehicle" subtask. (**Right**) Unstructured pruning for different models.

In the experiments seen in Figure 6 and in the Supplementary, we take categories on different levels of the taxonomic hierarchy, and compare performance between a sparsity solver applied on the whole iNaturalist dataset vs task-aware compression. The setting is the same as for ImageNet: unstructured sparsity, and fine-tuning via the L2 distillation strategy.

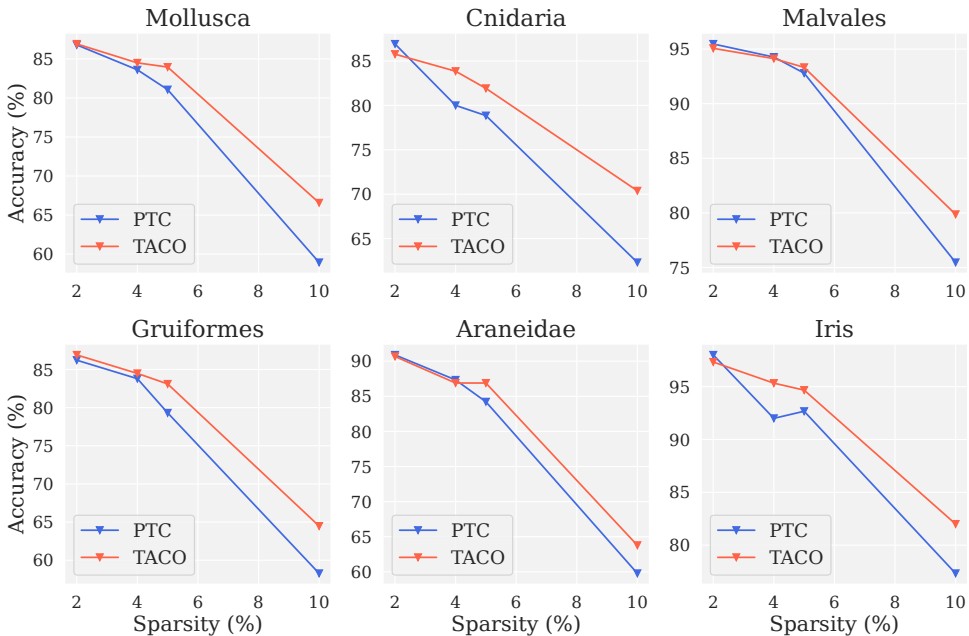

Figure 6: Few-shot compression on iNaturalist21

The results for 6 subtasks with 10-100 classes are shown in Figure 6, showing that the impact of task-aware compression on model performance is pronounced, especially with increasing sparsity. See Supplementary E for more details.

### 3.4 Speedups

Next, we examine the practical potential of our method both for CPU and GPU inference. We measure the execution latency for processing a single image of size $224 \times 224$ with a DeiT-III-Base model. (We note that larger batch processing leads to higher speedups, but would be less realistic in our target deployment scenario.) Full details about the benchmarking environment are provided in Supplementary D.

**CPU Speedups.** For experiments with CPU speedups we execute our unstructured sparse models on the DeepSparse runtime (Kurtz et al., 2020), which can accelerate unstructured sparsity. Latency is measured on an Apple M1 processor, using 4 cores. We also quantize both model weights and activations to 8 bits, which can be done losslessly. In Figure 7 (**left**), we observe that one can reduce latency by more than 4x with a minor performance drop (2%), and by 6x with a moderate (6%) accuracy drop.

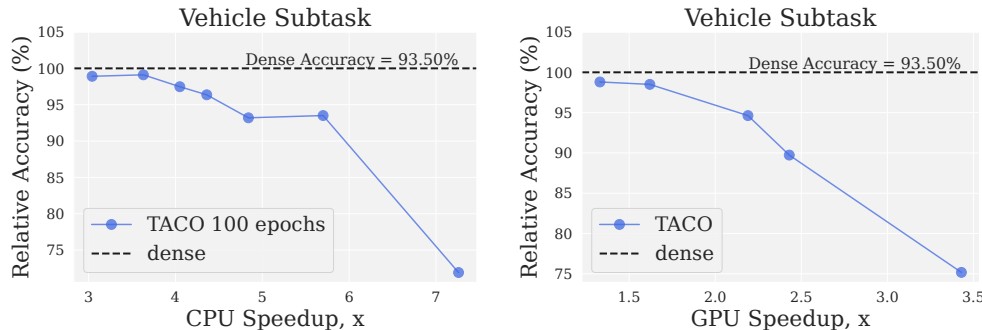

Figure 7: (**Left**) CPU speedups for DeiT-III-B vs relative accuracy with 8-bit quantization on the DeepSparse runtime. (**Right**) GPU speedups for DeiT-III-B vs relative accuracy with fp16 precision on the TensorRT runtime.

**GPU Speedup.** Next, we execute the structured-pruned models obtained via TACO (Section 3.2.2) in the NVIDIA TensorRT runtime, on an NVIDIA T4 GPU. We measure speedups and accuracies when specializing on specific subtasks, shown in Figure 7 (**right**). The speedups are identical on any subtask, as the classification head has a negligible cost. We observe that, through specialization, we obtain a 60% inference speedup with negligible accuracy loss, and 2.5x speedup while maintaining $\geqslant 90\%$ of the accuracy on this particular task and model combination.

### 3.5 Ablation Studies

**Solver Impact.** We have focused our experiments so far on the HybridOBC solver as the main method providing a trade-off between performance and efficiency in the few-shot scenario. In this section, we justify this choice by comparing with alternatives. Specifically, Figure 8 (**left**) shows that data-agnostic Magnitude pruning falls behind data-aware solvers, which leverage calibration data. Here, HybridOBC is more accurate than FastOBC, and performs on par with OBC, while being significantly faster. Specifically, on DeiT-III-Base, Fast OBC takes 12 second, while HybridOBC takes around 12 minutes, and OBC takes more than one hour.

**Data Impact.** The behavior of our proposed TACO approach with respect to the number of samples per class and calibration data fine-tuning passes is given in Figure 9. Naturally, accuracy improves with more data, but also shows diminishing returns. We find that 10 samples per class is a good balance between accuracy and speed of the approach.

Further, when a large amount of data is available, one can enhance the TACO effect by employing a more data- and compute-intensive Fisher-based approximation of the Hessian, together with longer fine-tuning. See Figure 10 for a comparison with the Fisher-based WoodFisher Singh & Alistarh (2020) and CAP Kuznedelev et al. (2024) pruners, and Appendix C for full results. However, in the limited-data regime with 10 samples per class, the fast HybridOBC solver works best.

**Distillation Impact.** Another ingredient we examine is the distillation strategy. The simplest approach would optimize the cross-entropy task loss on the calibration data; yet, this option results in suboptimal convergence as shown in Figure 8 (**right**). We find that, since there is only a scarce amount of data available, the model simply memorizes the samples instead of generalizing on the task. Output distillation reduces overfitting to the calibration set, but does not facilitate fast recovery. The compound loss we propose,

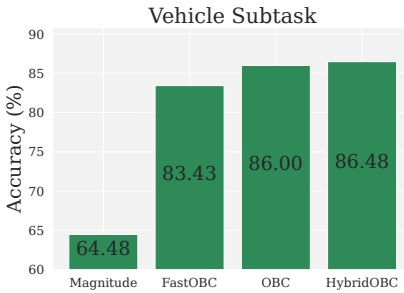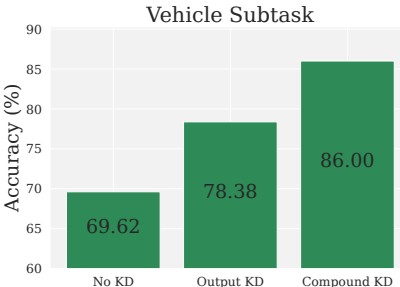

Figure 8: (**Left**) Ablation of pruner choice. (**Right**) Ablation of distillation strategy.

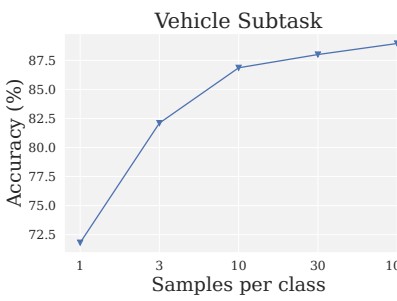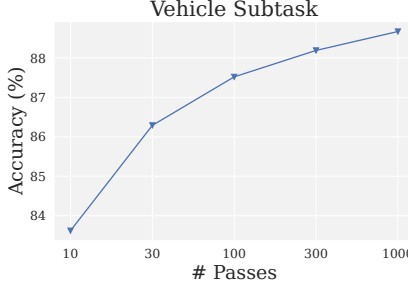

Figure 9: (**Left**) Accuracy on Vehicle task vs number of samples per class for fixed number of training steps for 90% sparse DeiT-III-Base. (**Right**) Accuracy vs number of passes through few-shot training set for 90% sparse DeiT-III-Base.

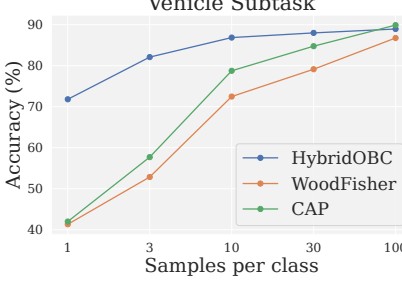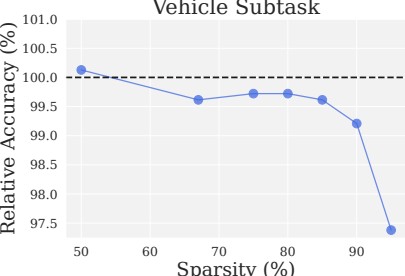

Figure 10: (**Left**) Accuracy on Vehicle task vs number of samples per class for different pruners. (**Right**) Accuracy on Vehicle subtask for CAP pruner in data-rich regime.

involving both the task loss, output logit distillation and intermediate activation representation, appears to noticeably improve upon the former two options. The evolution of validation accuracy and training loss is presented in Figure 11.

## 3.6 TACO on transfer learning tasks

The next test case of the introduced "TACO effect" is a transfer learning setting. Since ImageNet pretrained models are known to provide a good initialization for "downstream" tasks (Yosinski et al., 2014), we consider transfer to 6 tasks - CIFAR-10, CIFAR-100, Caltech-101, Caltech-256, Oxford Pets, and Flowers 101. Firstly, we tune a dense, ImageNet-pretrained model on the task of interest. Then, we prune the resulting model to the target compression rate, and fine-tune it via the calibration set, using the model from the previous step as a teacher. As before, we use 10 samples per each class and 100 passes through the calibration set. The PTC variant uses generic ImageNet-1k data for calibration, whereas TACO uses calibration data from the downstream task. We wish to determine whether TACO leads to a better compression / accuracy trade-off.

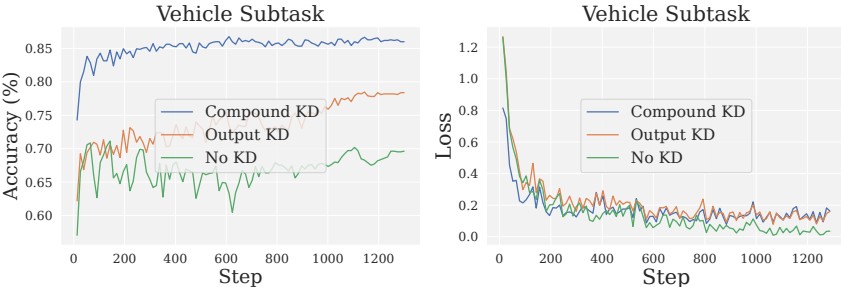

Figure 11: (**Left**) Validation accuracy vs step for different distillation options. (**Right**) Train loss accuracy vs step for different distillation options

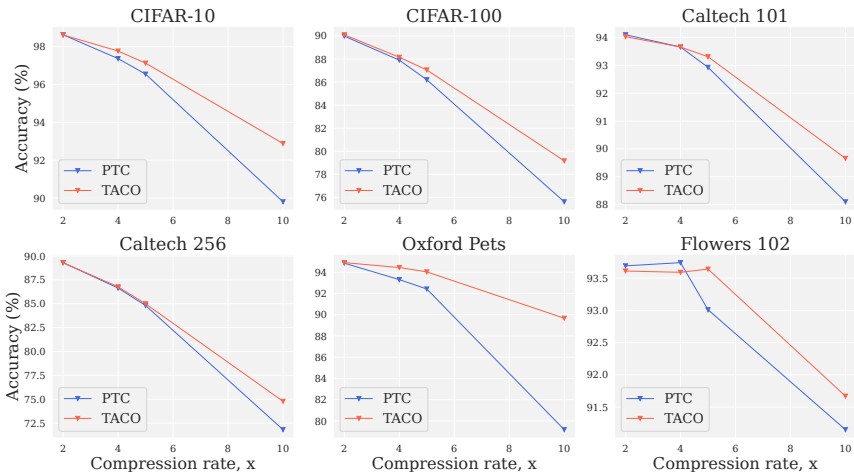

Figure 12: Few-shot compression on transfer learning tasks

The results in Figure 12 confirm that TACO achieves better performance than PTC in most cases. (On the X axis, the compression rate is defined as $\frac{1}{1-s}$ for a sparsity $s$.)

### 3.7 Gradual Pruning and Data-Aware Lottery Tickets

So far, our approach has been to apply TACO *one-shot*, initially, followed by fine-tuning. We next investigate its application *gradually*, in steps of increasing sparsity, separated by short fine-tuning. This gradual approach, formalized in Algorithm 1 coincides with Lottery Tickets for transfer learning as proposed by Chen et al. (2021b), and thus provides a good point of comparison relative to this method.

---

**Algorithm 1** Overview of gradual transfer learning setup.

---

1: **Input**: Dense model $M$, Sparsity target $\sigma$, Pruning method $\Gamma$.
2: **repeat**
3:     Prune 50% of remaining weights in $M$ via method $\Gamma$.
4:     Update $M$ to the pruned model.
5:     Finetune $M$ for 25 epochs, maintaining the sparsity mask.
6: **until** $M$'s sparsity reaches some target level $\sigma$.
7: Return $M$.

---

Following Chen et al. (2021b), we use a pretrained ResNet50/ImageNet-1K model as the base, and run TACO on a subset of 1024 samples randomly selected from the transfer task. In the gradual case, each pruning step removes 50% of the remaining model weights, followed by 25 epochs of full fine-tuning on the transfer task. In the single-step case, we prune once, and fine-tune for the cumulative equivalent number of epochs. For fairness, we use exactly the same setting with gradual magnitude pruning (GMP) and Lottery Tickets for

Transfer (LTH) Chen et al. (2021b) methods, applying the corresponding pruner at the compression step. Figure 13 shows results for 6 standard transfer datasets Kornblith et al. (2019), for compression between 2 (50% sparsity) and 64 (98.5% sparsity). We provide dedicated plots in Appendix A.

Across all tasks and sparsities, Gradual TACO achieves highest accuracy, followed by single-step TACO + finetuning. Remarkably, Gradual TACO achieves moderate accuracy loss even for $\geq 95\%$ sparsity, outperforming the GMP and LTH data-agnostic methods by a large margin. The strong performance of Single-step TACO shows that specialization can occur even when there is just a single compression step, and the calibration data is *not* part of the pretraining task. This suggests that model specialization may also happen for tasks that are not strictly in-distribution, which bodes well for the generality of the method. The accuracy difference between TACO and LTH/GMP shows that task-aware specialization provides much better compression for transfer learning.

## 4 Application to text-to-image generation

TACO can be applied to any deep-learning based computer vision task where there is potential for specialization. To illustrate this, we show results for text-to-image generation on the Pokémon BLIP captions dataset Pinkney (2022) using a Stable Diffusion (SD) v1.4 model trained via the Hugging Face `diffusers` tutorial[1].

We prune convolutions and linear projections in the SD UNet while keeping the image encoder dense. Since the SD UNet is much larger than ViT models considered in previous experiments ($\sim$860M parameters) we apply the FastOBC sparsity solver. We use the whole task data, consisting of $\sim$ 800 images, as a calibration set, placing us in the few-shot tuning regime. We finetune the single-step compressed model for the same amount of steps used for teacher specialization.

To quantitatively assess image generation quality, we employ the commonly used FID (Fréchet Inception Distance) metric (Heusel et al., 2018), which utilizes the Fréchet distance between features extracted by the Inception-V3 (Szegedy et al., 2016) backbone pretrained on ImageNet. This metric measures the distance between the target and generated data distributions. Specifically, using prompts from the Pokémon dataset, we generate images through dense and sparse diffusion models and calculate the FID metric based on statistics from both the original dataset and the generated images. The results are presented in Table 1.

It can be observed that the FID score for the model with 50% sparsity is similar to that of the original model. The increase in the FID score remains moderate at 67% and 75% sparsity levels. Additionally, we provide qualitative results for eight random prompts (see Appendix G for details) in Figure 17.

Table 1: FID score for Stable Diffusion v1.4 models with different sparsity.

| Sparsity (%) | FID |
| --- | --- |
| 0 | 56.91 |
| 50 | 53.96 |
| 67 | 61.45 |
| 75 | 63.05 |

## 5 Discussion

In this paper, we describe and analyze for the first time the effect of task specialization in the context of model compression. Via systematic experiments, we show that task-specificity can increase the degree of compression by roughly 2x at the same level of accuracy, relative to task-agnostic techniques. Our findings are valid across one-shot compression, fine-tuning, transfer learning, and gradual pruning scenarios. They lead to significant speedups (of up to 6x) at low to moderate accuracy loss, relative to the original model, which applies to both commodity CPU and accelerator GPU hardware.

A strength of our approach is that it is generic with respect to the sparsity solver used. We have shown notable differences in performance with respect to different solvers, in particular relative to the amount of data and computation employed. Future work could investigate specialized solvers for task-aware compression, or applications to other domains, such as detection or segmentation.

---

[1] https://huggingface.co/docs/diffusers/training/text2image

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

# A    Transfer learning results

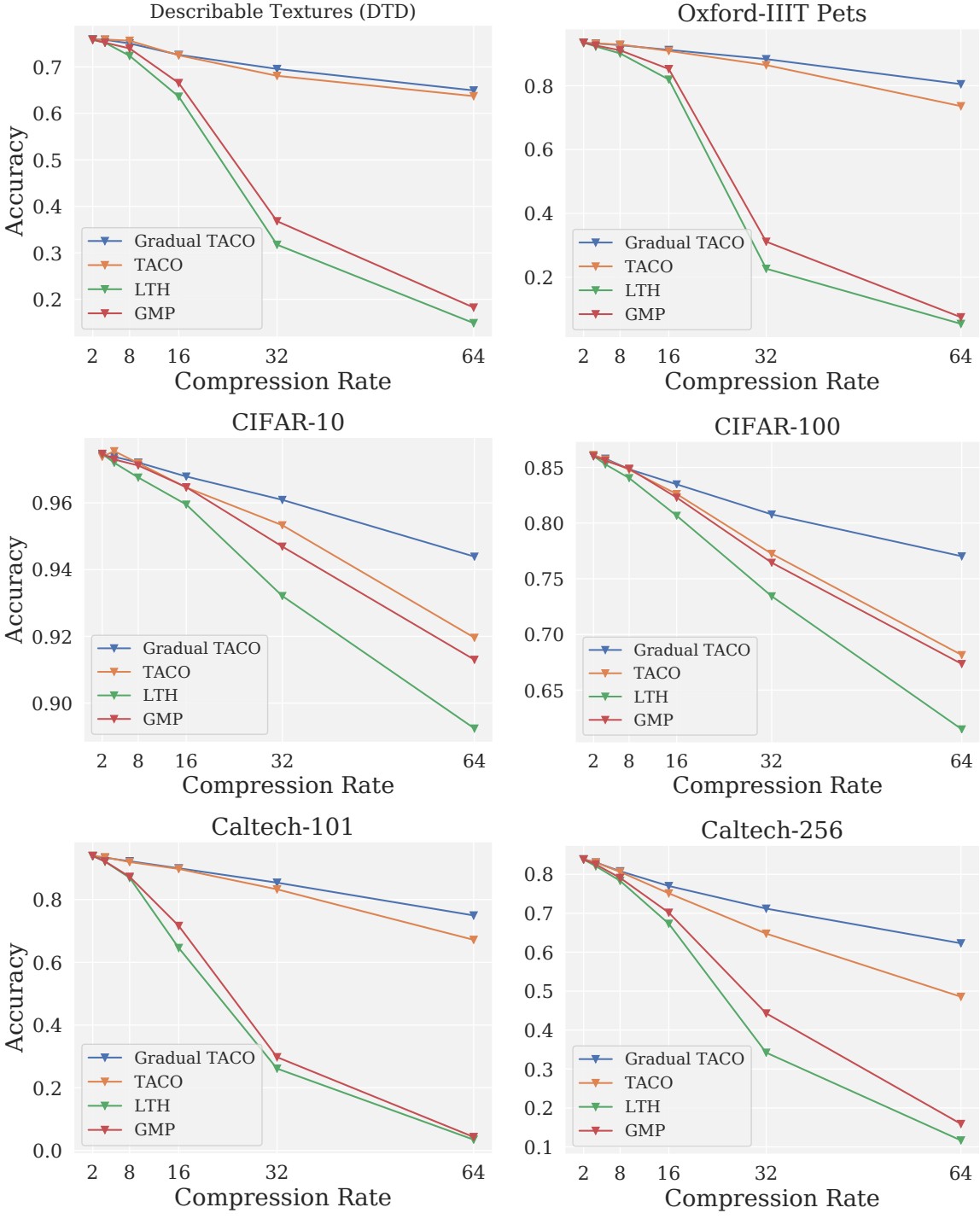

Figure 13: Experiments for model compression in a standard transfer learning setup. We compare TACO and Gradual TACO with Lottery Tickets for Transfer (LTH) and Gradual Magnitude Pruning (GMP) in terms of accuracy (Y axis) vs compression (X-axis).

## B  Few-shot tuning hyperparameters

In our few-shot tuning procedure we finetune the model via optimizing the sum of original task loss, logit and feature distillation loss. The hyperparameters adopted in training are listed in Table 2. Learning rate has been determined via grid search on $\{1 \cdot 10^{-5}, 3 \cdot 10^{-5}, 1 \cdot 10^{-4}, 3 \cdot 10^{-4}, 1 \cdot 10^{-3}, 3 \cdot 10^{-3}\}$ on "Vehicle" subtask and adopted in all other experiments. We apply simple augmentation pipeline Szegedy et al. (2014) with random resized crops and horizontal flips. Since distillation with intermediate teacher activations is already a strong regularizer there is no need for an elaborate augmentation/regularization pipeline.

Table 2: Hyperparameters used in few-shot tuning.

| | |
|---|---|
| Optimizer | Adam |
| Learning rate | $3 \cdot 10^{-4}$ |
| LR schedule | linear |
| Batch size | 16 (32) |
| Num passes | 100 |
| Weight decay | 0 |
| Dropout | ✗ |
| H.flip | ✓ |
| RRC | ✓ |
| Image size | 224 |
| Test crop ratio | 0.875 |

Losses forming the compound distillation loss may differ significantly in magnitude and the importance for generalization performance. While the identification of optimal weighting of the losses can improve performance, we decided to keep our setup simple and tried couple values $\lambda_{feat}$ in front of feature loss. It turned out that the results are robust to its variation within a wide range of values. However, if the coefficient in front of feature loss is too small, guidance from teacher activations may be not sufficiently strong.

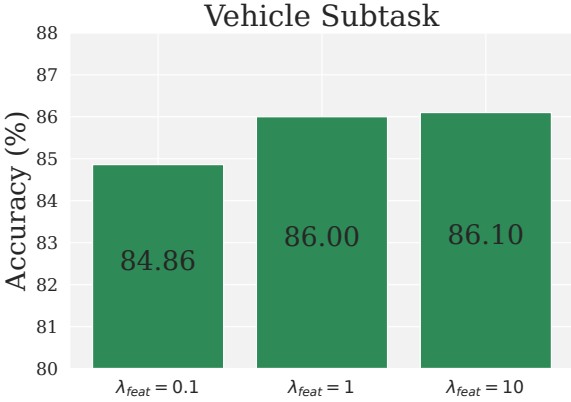

Figure 14: Ablation of the coefficient in front of feature loss.

## C  Performance of Fisher-based pruners

The two most commonly applied approximations of the Hessian for estimates of parameter importance are the layerwise approximation Dong et al. (2017); Frantar et al. (2022); Frantar & Alistarh (2023); Kurtic et al. (2023) and block-diagonal Fisher approximation Singh & Alistarh (2020); Frantar et al. (2021); Kuznedelev et al. (2024). The first one accounts only for importance of specific weight (group of weights) on the output

of given layer, whereas the latter estimates the saliency with respect to the target task. However, it appears that methods leveraging Fisher approximation demand more samples for accurate estimate and fall behind layerwise sparsity solvers for small number of samples (Figure 15 (**Left**)). With the increase of amount of data for Empirical Fisher WoodFisher and CAP close the performance gap with HybridOBC and can even outperform it.

To test our method in data- and compute- rich regime we took 200 samples (20% of the training samples from ImageNet-1k) per class for Vehicle subtask and DeiT-III-Base model and finetuned the model for 100 passes through the calibration set. One can preserve 97.5% of the original model performance even at the highest 95% sparsity considered. Therefore, the proposed method works well even in the data abundant setup.

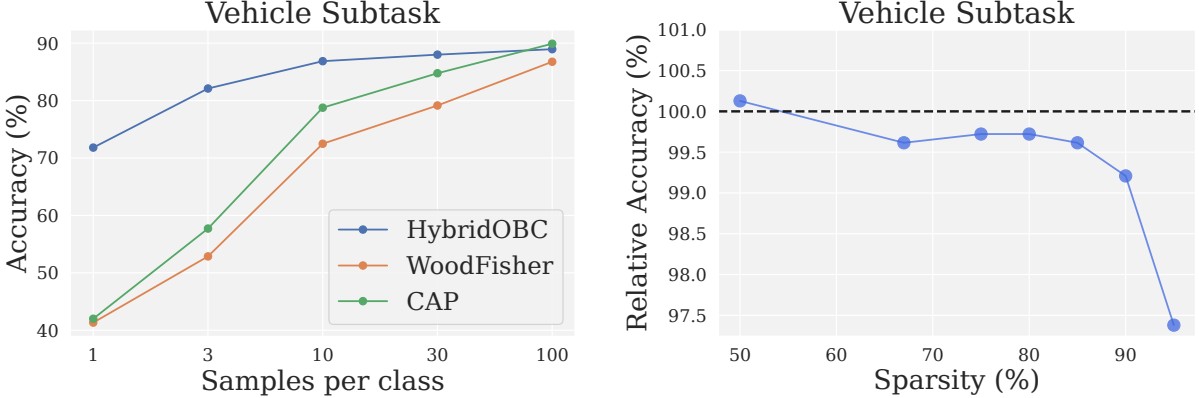

Figure 15: Accuracy on Vehicle subtask for CAP pruner.

## D    Benchmarking environment

**CPU**    benchmarking was conducted on 4 cores of Apple M1 chip (MacBook Pro 2020). Model was compiled with DeepSparse engine (version `deepsparse-nightly 1.6.0`).

**GPU**    measurements were carried out on Google Colab runtime, Nvidia T4 GPU, TensorRT (version `8.6.0.12` for CUDA Toolkit 12.0).

## E    Details about iNaturalist subtasks

In the main text we present results for 6 tasks from different levels of species taxonomy. Below are presented some additional details about the chosen tasks.

Table 3: Additional details about iNaturalist subtasks

| Dataset | Category | Classes |
|---------|----------|---------|
| Mollusca | phylum | 169 |
| Cnidaria | phylum | 26 |
| Malvales | order | 75 |
| Gruiformes | order | 29 |
| Araneidae | family | 45 |
| Iris | genus | 15 |

## F  TACO vs PTC on the ImageNet hierarchy

In order to demonstrate that TACO superiority is not caused by cherry-picking of specific ImageNet subsets we evaluated TACO and PTC compression on the whole set of ImageNet subtasks with more than 5 classes (i.e. 166 subtasks chosen). Comparison plots are presented on Figure 16. One can observe that TACO outperforms PTC on most of the tasks.

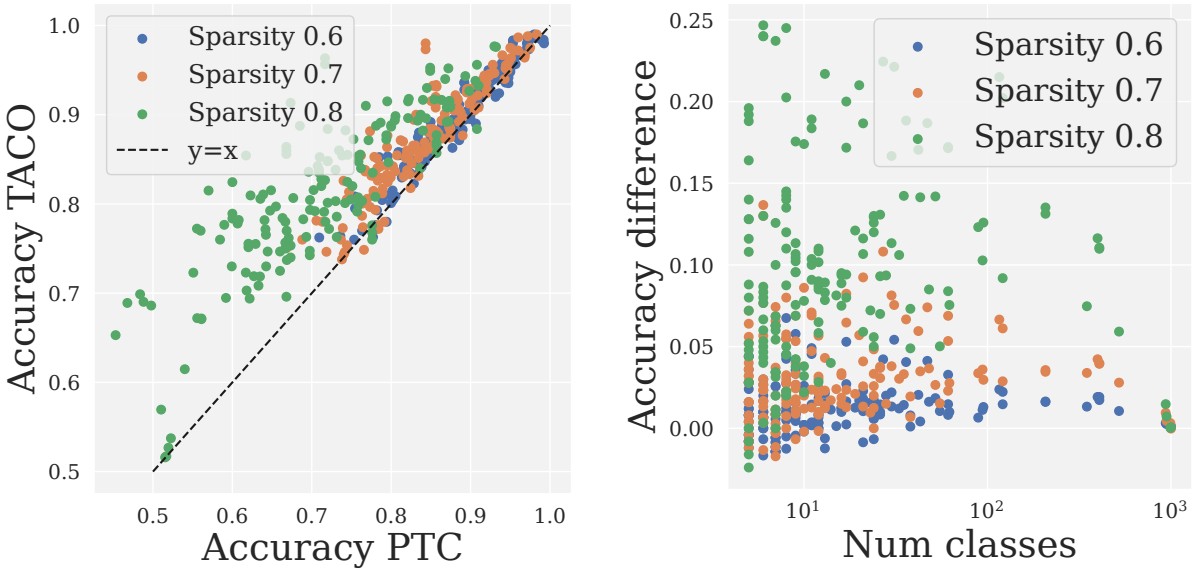

Figure 16: (**Left**) Accuracy of PTC compressed model vs TACO on the specialized task (ImageNet subset). (**Right**) Accuracy difference between models compressed by TACO and PTC on the specialized tasks.

## G  Prompts and visualization in text-to-image experiment

The prompts used for generation of images 17 are the following:

- a cartoon bear with a ring around its neck

- a very cute looking cartoon character with big eyes

- a cartoon character is holding a baseball bat

- a drawing of a green and white bird

- a bird with yellow eyes and a black beak

- a drawing of a white and black animal with blue eyes

- a drawing of a dragon with its mouth open

- a cartoon picture of a giant turtle with its mouth open

Images were generated with default SD solver, 50 inference steps and cfg scale of 7.5.

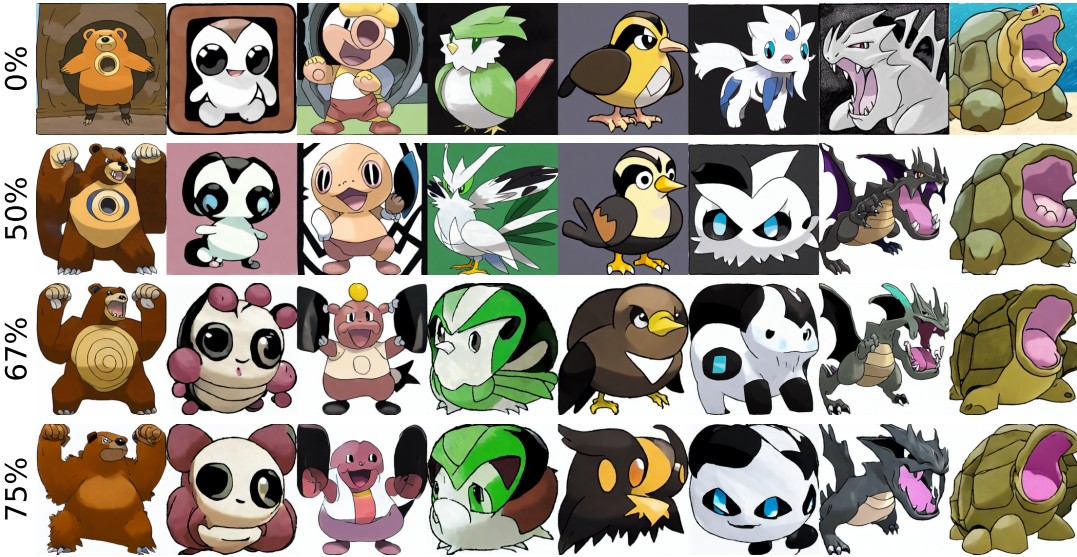

Figure 17: Generated images corresponding to random prompts from Pokémon BLIP captions. Percentages are sparsity levels.

## H Sensitivity to the choice of random seed

The experiments in the main part of the text adopt single choice of random subset for specific subset. To verify the robustness of the results to the choice of random seeds we repeat the experiments for vehicle, dog and spider subtasks from ImageNet in single-step compression setting for 4 random seeds both for TACO and PTC and observed that the final result is not very sensitive to the choice of random seed with TACO exhibiting smaller variance compared to PTC. Results averaged across multiple runs are provided on Figure 18.

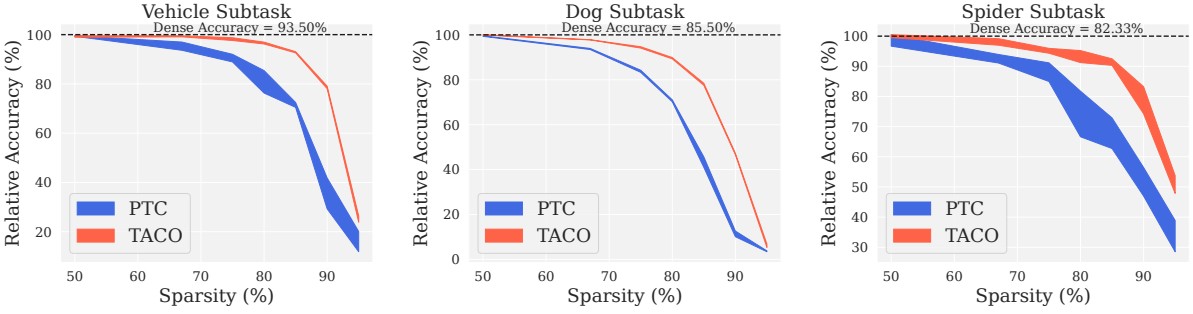

Figure 18: Mean and standard deviation for single-step compression on ImageNet.

