# OpenReview forum: "TACO Vision Models Can Be Efficiently Specialized via Few-Shot Task-Aware Compression"
_TMLR — Accepted by TMLR_

### Review · Reviewer_rnwv · 2024-12-16

**Summary Of Contributions:**

The paper proposes Few-Shot Task-Aware Compression (TACO), a method to compress large generalist vision models into efficient task-specific models using a small amount of calibration data. The approach combines task-specific pruning and few-shot fine-tuning with knowledge distillation to maintain accuracy while achieving significant parameter reduction and computational efficiency. The authors validate their method across multiple vision architectures (e.g., ResNet, ViT, ConvNeXt) and datasets (ImageNet, iNaturalist), demonstrating the effectiveness of the approach.
The paper presents an interesting and relevant approach to task-aware model compression. However, issues with the presentation and missing contextualization with related work weaken the overall quality of the paper.

**Audience:**

Yes

**Claims And Evidence:**

No

**Requested Changes:**

I expect the four identified weaknesses to be thoroughly addressed, at which point I will reassess my evaluation. Specifically, the paper requires the inclusion of a related work section, comprehensive comparisons with prior approaches, reporting of standard deviation, and a fundamental revision of its formatting.

**Strengths And Weaknesses:**

Strengths:

S1. The topic is relevant, well-motivated, and clearly presented, making the paper easy to follow in terms of writing.

S2. The proposed method is straightforward and easy to understand.

S3. The quantitative results are promising. The authors demonstrate the method's efficacy across multiple model architectures and datasets.


Weaknesses:

W1. Missing Related Work Section -- The related work section is absent. Given that the paper addresses multiple dimensions, it is crucial to situate the proposed approach within the context of prior research. There is substantial existing work on both few-shot learning and model compression. A detailed comparison and discussion of related approaches are necessary.

W2. Lack of Experimental Comparison with Prior Work -- While the method demonstrates promising quantitative results favorable compared to the proposed Post-Training Compression (PTC) baseline, comparisons with prior work or baselines constructed using existing methods are missing. Such comparisons are essential for contextualizing the contribution of this work.

W3. Sample Dependence and Lack of Error Bars -- The method relies on a small number of samples (10 images) to perform task-aware compression. This raises concerns about the method's dependence on the chosen samples. Reporting error bars in the plot visualizations is critical to account for variability and enhance the reliability of the results.

W4. Severe Formatting Issues -- The paper suffers from formatting issues, particularly in the figures and references.
- Section headings (e.g., "Method") are placed too close to the preceding text, rather than the content they introduce.
- Figures are poorly formatted. For example, Figures 2 and 5 have captions that are cryptic or insufficiently detailed, making interpretation difficult without checking the text in detail.
- Figures lack consistency in width, font size, and alignment (e.g., Figure 2 vs. Figure 3).
- Legends overlap with content despite there being white space available (e.g., Figure 4).
- References are inconsistent in venue naming: for instance, Hubara et al. (2022) use "NeurIPS," whereas Chen et al. (2021a) use "Advances in Neural Information Processing Systems," and Dong et al. (2017) use "Conference on Neural Information Processing Systems (NeurIPS)."
- Some references (e.g., Szegedy et al., 2015) lack venue information.
- Capitalization of paper titles and inclusion of page numbers are inconsistent.


Minor Remarks:
- Abstract: The abstract is overly long and challenging to parse. It could be shortened and made more precise to align with the paper's overall length.
- Typo in Section 4: The phrase "with via" appears in the second sentence of "Application to Text-to-Image Generation."
- Figure 11: The legend is positioned in the middle of the plot. Consolidating legends into a single, well-placed instance (e.g., in the plot on the right) would improve clarity.
- Figures 13 and 14: These figures appear before the first section of the supplemental material, disrupting the logical flow.

---

> ### Author Response · Authors · 2025-01-10
>
> We thank the reviewer for thorough and detailed review. The concerns are addressed below:
>
> W1. To the best of our knowledge, we are the first to study the problem of task-specific compression, rather than preservation of generic abilities under compression. Few-shot learning is an interesting and rich topic, but it is irrelevant to the current study. In the original version of the manuscript, we provided a brief overview of the model compression problem in the introduction. According to the suggestion of the reviewer, we added a dedicated related work section about model compression.
>
> W2. As discussed above, we are unaware about any other work studying task-specific compression. Therefore, PTC is the only natural baseline to TACO.
>
> W3. We added a dedicated ablation study for ImageNet subtasks to Appendix H. Overall, our observations indicate that the proposed method is robust to the choice of samples, and the improvement of TACO over PTC seems to be statistically significant.
>
> W4. We appreciate the reviewer's thorough evaluation of the manuscript. We have revised the paper in line with the suggested changes.

---

### Review · Reviewer_rVaw · 2024-12-19

**Summary Of Contributions:**

This paper introduces TACO, a method for efficiently specializing pre-trained computer vision models for specific tasks using very small amount of task-specific data. A novel two step process was proposed, combing task-specific compression and KD based fine-tuning. The authors evaluate the proposed method in various network architectures and datasets, and further extend the evaluation task from classification to image generation. Results show TACO achieves 2x compression compared to task-agnostic method, and providing significant computational speedups both in CPU and GPU.

**Audience:**

Yes

**Broader Impact Concerns:**

No concerns.

**Claims And Evidence:**

No

**Requested Changes:**

1. Please indicate what method used in Figure 2 in the caption. Is it two-step TACO?
2. For the text-to-image generation, it is better to consider adding quantitative result for assessment. I suggest add another dataset that can be evaluate by FID metric.
3. There are some repeat citations, like ViTs are cited twice in the same way in the introduction section.
4. The equation $f_C^D=TACO(f,S,C,D)$ is used for defining the input of TACO but seems express same information as the text above and not relevant to other equations below.

**Strengths And Weaknesses:**

Strengths:
1. Authors evaluate TACO in various settings and datasets. The extensive experiments on multiple datasets, models and subtasks establish the generality and efficacy of TACO.
2. The task-aware compression approach is novel and demonstrates potential for real-world applications in resource-constrained scenarios like identifying species in the wild.
3. The paper has detailed ablations, quantitative and qualitative analysis to strengthen the claims.

Weaknesses:
1. In the main experiments, authors use 10 samples per class for the calibration data but failed to discuss the impact of the number of class used for the calibration dataset.
2. This paper focuses primarily on vision classification tasks and extend a bit to image generation task, but only report a naive visualization result in supplementary material for image generation quality, without any quantitative analysis.

---

> ### Author Response · Authors · 2025-01-10
>
> We thank the reviewer for feedback. We address the concerns below:
>
> 1. Figure 2 highlights the rationale for transitioning from task-generic compression—due to the challenge of maintaining accuracy uniformly across all domains—to task-specific compression. We apologize for any confusion and have explicitly noted in the caption that Figure 2 relates to PTC (generic compression).
>
> 2. We agree with the reviewer that quantitative assessment is crucial to validate the efficacy of the proposed approach. Following the suggestion, we calculated the FID (Fréchet Inception Distance) for the entire training set of the Pokémon BLIP captions dataset, as well as for datasets generated using the same captions through both dense and sparse Stable-Diffusion v1.4 models. The results, which have been included in the revised manuscript, are presented below:
>
> | Sparsity  (%) |  FID  |
> |:-------------:|:-----:|
> |       0       | 56.91 |
> |       50      | 53.96 |
> |       67      | 61.45 |
> |       75      | 63.05 |
>
> 3. It can be observed that the FID score for the model compressed to 50% sparsity is similar to that of the original model. The increase in the FID score remains moderate even at 67% and 75% sparsity levels.
>
> 4. Thank you for noticing that. We fixed this in the updated revision.
> The purpose to define TACO in a form of equation is for better readability and putting emphasis on the main idea. We agree that the equation for TACO represents the same idea as the paragraph above.

---

### Review · Reviewer_e5gK · 2024-12-28

**Summary Of Contributions:**

Few-Shot Task-Aware COmpression (TACO).

In this paper, we seek to extend the reach of post-training compression to consider the requirement of model specialization. We propose a simple and versatile approach which we call Task-Aware COmpression (TACO), illustrated in Figure 1. Given a pre-trained generalist model and a “calibration set” containing a few samples for categories of interest, TACO can produce an efficient specialist model in two
steps.

There are two main steps in TACO including the model compression and training on the representative training images. In my view, model compression makes the most esstial novel role in the novelty of the TACO.

Step1. Author split the network layer-wise, and run task-specific calibration data D through the uncompressed model. Compression is then applied to each layer by solving a layer-wise optimization problem. We also propose HybridOBC, a solver which can be seen as a best-of-both-worlds composition of OBC [1] and FastOBC [2].

Step2. In order to get rid of the negative effect on the accuracy caused by the compression, the author employ a limited few-shot form of finetuning to re-calibrate the model and recover additional accuracy. We address both challenges by introducing a self-distillation-based approach where they user a self-distillation-based approach [3] to recover the accuracy of the sparsified model relative to the dense one, which does not overfit when applied over a small calibration set.

TACO models provide end-to-end computational speedups 1.5-3x relative to the uncompressed models, at similar accuracy levels to dense, and 3-6x at moderate accuracy loss, on both CPUs and GPUs.


[1] Frantar E, Alistarh D. Optimal brain compression: A framework for accurate post-training quantization and pruning
[2] Frantar E, Alistarh D. Sparsegpt: Massive language models can be accurately pruned in one-shot[C]//International Conference on Machine Learning. PMLR, 2023: 10323-10337
[3] Zhang L, Bao C, Ma K. Self-distillation: Towards efficient and compact neural networks[J]. IEEE Transactions on Pattern Analysis and Machine Intelligence, 2021, 44(8): 4388-4403.

**Audience:**

Yes

**Claims And Evidence:**

Yes

**Requested Changes:**

Same as above.

**Strengths And Weaknesses:**

Strengths:
1. The introduction of HybridOBC seems reasonable and extensive experiments illustrate the effectiveness of the work.
2. This paper is well-organized and easy to read.
3. This paper has done extensive experiments in verifying the efficency of the work.

Weakness:
1. According to many recent work which aims in adaping the large pre-trained models such as prompt tuning in CLIP[1], it aims in well-adapting on the downstream tasks suchas cars while maintaing the well generailized ability. If the aim is to erasing the ability of it, why not train a model from scratch. If so, there is no experiment comparing it with train from scratch.

2. During the experiment, nearly all the experiments or figures are based on the TACO while there is no comparison between existing models which make the paper less convincing.


[1] Tian X, Zou S, Yang Z, et al. ArGue: Attribute-Guided Prompt Tuning for Vision-Language Models[C]//Proceedings of the IEEE/CVF Conference on Computer Vision and Pattern Recognition. 2024: 28578-28587.

---

> ### Author Response · Authors · 2025-01-10
>
> We thank the reviewer for insightful review and comments. We address the concerns below:
>
> 1. The primary aim of our work is not to develop new abilities, but to maintain the specific skills that the model already possesses. Given a generalist model, trained on large and diverse dataset, we show that achieve high compression rates at small computational budget, if one focuses on preserving knowledge from narrow domain (for example model, capable of distinguishing between shark or turtle species) instead of trying to preserve quality uniformly across all possible classes. To accomplish this goal, we propose TACO, a method that utilizes task-specific data selection for both compression and fine-tuning processes. By few-shot compression we mean that a small amount of data is sufficient to produce good results.
>
> 2. In our paper, we compare post-training compression (PTC), which involves using generic data sampled uniformly across a large dataset, with TACO, which focuses on task-specific calibration. We validate our approach on different tasks, models and datasets and the comparison makes sense only across a single model and task (due to different number of classes in different tasks and baseline accuracy) and show that TACO performs consistently better on task-specific compression.

---

### Author Response · Authors · 2025-01-10
**Summary of changes and general comment**

We thank the reviewers for their insightful responses.

Below we provide summary of changes made based on the received feedback:

* Added quantitative assessment (FID score) of image generation quality (Reviewer rVaw) for dense and compressed diffusion generative models.
* Added ablation (**Appendix H**) with respect to randomness to estimate the statistical significance of the paper claims and variance in the experiments (**Reviewer rnwv**). Overall, the obtained results suggest that the conclusions made are robust to random seeds and the advantage of TACO over PTC is statistically significant.
* Restructured some parts of the paper and figure placement according to the suggestions of **Reviewers rVaw and rnwv**. References have been fixed.

We would like to emphasize that the goal of the paper is efficient and accurate compression of the generalist model while focusing on the specific ability of the model. Few-shot learning, despite being a rich and interesting topic, is not directly related to our study as its target is quick adaptation of some model to new tasks, whereas in our approach we aim to preserve knowledge already present in the model.

---

### Decision · Action_Editor_22se · 2025-02-07

**Recommendation:** Accept with minor revision

**Comment:**

The paper has the potential to be a valuable contribution to TMLR by introducing a novel and practical approach to task-aware model compression. However, it currently suffers from weaknesses in evidentiary support, benchmarking, and presentation, as highlighted by the reviewers.  "Accept with Revision" is justified because the core idea is promising, and the identified weaknesses are addressable through focused revisions.  Addressing these points will significantly strengthen the paper and make it a more convincing and impactful contribution to the field.  If the authors can successfully address these revision requirements, the paper will be a candidate for acceptance in TMLR.

**Audience:**

Some individuals within TMLR's broad audience would be interested in knowing the findings of this paper.

**Claims And Evidence:**

The reviewers have mixed opinions on the strength of the evidence. Their detailed review points out sample dependence and lack of error bars (W3). Another concern is the lack of comparison with existing models. Reviewer rVaw pointed out a weakness related to the number of calibration classes and the lack of quantitative analysis for image generation. Reviewer rnwv's official recommendation is "Leaning Reject", citing that the rebuttal did not adequately address their concerns. The limited comparison to existing methods and baselines (e5gK, rVaw's initial comment) prevents a clear understanding of TACO's advantages and disadvantages compared to the broader landscape of model compression and specialization techniques.

---

> ### Author Response · Authors · 2025-02-24
> **Official comment to Action Editor**
>
> We thank the reviewers and action editor for their thoughtful reviews and valuable suggestions! Based on the received feedback, we introduced following changes to the submission manuscript:
>
> 1. To address the questions regarding statistical significance, we have added a dedicated ablation section in the Appendix (Appendix H) on the impact of the random seed. The results show that the advantage of TACO over PTC is statistically significant.
>
> 2. Next, we address the question of comparison with few-shot learning.
> Here, we wish to emphasize that, while few-shot learning is somewhat related to our work, it focuses on a different problem: adapting a generalist (pre-trained) model to a specific task, for instance by fine-tuning it on a few samples from that class.
> By contrast, our method focuses on compression by tailoring the model to extract only the specific knowledge already-present in the pre-trained model, relative to sub-tasks that the model was already able to solve.
> Since the objectives are different, an apples-to-apples comparison between our approach and few-shot learning is not possible. To address this, we have added a discussion about few-shot learning methods to a dedicated paragraph in the introduction, where we highlight the difference from the problem addressed in our work to few-shot learning to avoid potential confusion.
>
> 3. Finally, regarding scenarios with different number of classes, we note that we considered these already in the original paper revision. Specifically, the results presented on Figure 4 suggest that the compression task becomes harder, in general, with an increase of the number of classes. According to the suggestion of Reviewer rVaw, we added quantitative evaluation to the revised revision.
>
> We hope that these changes, incorporated into the revision, address the remaining questions and stay at your disposal for further inquiries.